# Are Sutureless and Rapid-Deployment Aortic Valves a Serious Alternative to TA-TAVI? A Matched-Pairs Analysis

**DOI:** 10.3390/jcm10143072

**Published:** 2021-07-12

**Authors:** Sameer Al-Maisary, Mina Farag, Willem Hendrik Te Gussinklo, Jamila Kremer, Sven T. Pleger, Florian Leuschner, Matthias Karck, Gabor Szabo, Rawa Arif

**Affiliations:** 1Department of Cardiac Surgery, University Hospital Heidelberg, 69120 Heidelberg, Germany; mina.farag@med.uni-heidelberg.de (M.F.); willemhendrik.gussinklo@med.uni-heidelberg.de (W.H.T.G.); jamila.kremer@med.uni-heidelberg.de (J.K.); matthias.karck@med.uni-heidelberg.de (M.K.); gabor.szabo@uk-halle.de (G.S.); rawa.arif@med.uni-heidelberg.de (R.A.); 2Department of Cardiology, Angiology and Pneumology, University Hospital Heidelberg, 69120 Heidelberg, Germany; sven.pleger@med.uni-heidelberg.de (S.T.P.); florian.leuschner@med.uni-heidelberg.de (F.L.); 3Department of Cardiac Surgery, Halle University, 06120 Halle, Germany

**Keywords:** heart valve, transapical, percutaneous (TAVI), aortic valve and root, outcomes (includes mortality and morbidity)

## Abstract

Background: Transcatheter aortic valve implantation is a feasible alternative to conventional aortic valve replacement with expanding indication extending to low-risk patients. Sutureless and rapid-deployment aortic valves were developed to decrease procedural risks in conventional treatment. This paired-match analysis aims to compare patients undergoing surgical transcatheter aortic valve implantation to sutureless and rapid-deployment aortic valve implantation. Methods: Retrospective database analysis between 2010 and 2016 revealed 214 patients undergoing transcatheter aortic valve implantation procedures through surgical access (predominantly transapical) and 62 sutureless and rapid-deployment aortic valve procedures including 26 patients in need of concomitant coronary artery bypass surgery. After matching, 52 pairs of patients were included and analyzed. Results: In-hospital death (5.8% vs. 3.8%; *p* = 0.308) was comparable between transcatheter aortic valve implantation (mean age 77 ± 4.3 years) and sutureless and rapid-deployment aortic valve implantation groups (mean age 75 ± 4.0 years), including 32 females in each group. The logistic EuroSCORE was similar (19 ± 12 vs. 17 ± 10; *p* = 0.257). Postoperative renal failure (*p* = 0.087) and cerebrovascular accidents (*p* = 0.315) were without significant difference. The incidence of complete heart block requiring permanent pacemaker treatment was relatively low for both groups (1.9% vs. 7.7%; *p* = 0.169) for TAVI and sutureless and rapid-deployment valves respectively. Intraoperative use of blood transfusion was higher in the sutureless and rapid-deployment aortic valve implantation group (0.72 U vs. 1.46 U, *p* = 0.014). Estimated survival calculated no significant difference between both groups after 6 months (transcatheter aortic valve implantation: 74 ± 8% vs. sutureless and rapid-deployment aortic valve implantation: 92 ± 5%; log rank *p* = 0.097). Conclusion: Since sutureless and rapid-deployment aortic valve implantation is as safe and effective as transapical transcatheter aortic valve implantation, combining the advantage of standard diseased-valve removal with shorter procedural times, sutureless and rapid-deployment aortic valve replacement may be considered as an alternative for patients with elevated operative risk considered to be in the “gray zone” between transcatheter aortic valve implantation and conventional surgery, especially if concomitant myocardial revascularization is required.

## 1. Introduction

Since the release of the latest European guidelines on cardiac valve disease and PARTNER 3 trial results, the controversy between conventional surgical aortic valve replacement and transcatheter aortic valve implantation (TAVI) is continuously fading in favor of TAVI, extending indication to intermediate- and low-risk patients [1,2]. Available data from randomized controlled trials have proven the non-inferiority and even superiority over surgery in this patient cohort if transfemoral access is suitable [3,4,5,6,7,8,9,10,11,12]. Future registry data on long-term outcomes after TAVI may enlighten this indistinct aspect.

However, surgeons strive to optimize conventional aortic valve replacement (AVR), which can first and foremost be achieved by reducing procedural and aortic cross-clamp time. Sutureless and rapid-deployment aortic valves (SURD-AVR) render the required premises by combining conventional surgical principles with easy implantation technique, leading to a hybrid-like procedure.

Moreover, these valve types facilitate minimally invasive AVR, which was limited in the past due to extended procedural times. Consecutively, several reports and meta-analyses compared AVR using sutureless and rapid-deployment aortic valves to TAVI [13,14,15,16,17,18,19,20,21,22]. The results led to the common conclusion supporting the use of SURD-AVR as an alternative to TAVI in high- or intermediate-risk patients requiring aortic valve replacement.

We aimed to compare the early outcomes after SURD-AVR and transapical (TA)-TAVI using a pair-match analysis. Transfemoral (TF) access is the first approach in our center. However, TF patients were excluded in our study, since TA access is the more invasive procedure requiring intubation of the patient. TA-TAVI patients generally present with higher morbidity and EuroSCORE values. This in turn provides better comparability to SURD-AVR in a real-life clinical setting, since SURD-AVR has been predominantly used in high-risk patients to reduce cross-clamping time. In addition to sutureless Perceval prostheses our cohort also included rapid deployment Intuity valves.

## 2. Methods

### 2.1. Study Population

214 TA-TAVI and 62 SURD-AVR, including 26 patients undergoing concomitant coronary artery bypass grafting (CABG) procedure, were identified between 2010–2016. After matching we collected the complete records of 52 patients in each group. Follow-up (mean 295 ± 450 d, median 196 d) was obtained through our institutional data base (only adult patients who were legally competent were included).

### 2.2. Surgical Procedures

SURD-AVR patients were operated on using extracorporeal circulation (ECC). Venous drainage was achieved either via bicaval or atrial cannulation. Activated clotting time (ACT) was aimed at 450 s by intraoperative heparinization before cannulation. A membrane oxygenator was applied, and surgery was performed at different levels of hypothermia (mean: 33 °C ± 1.8 °C) depending on the surgical procedure. Mean cross-clamp time was 64 min (±33 min). Five patients of the SURD-AVR group underwent access through partial sternotomy and 24 received concomitant CABG surgery, while 23 TAVI patients underwent PCI/stenting 6 weeks prior to surgery.

TAVI patients were operated on under full anesthesia using left anterolateral access. One patient underwent partial sternotomy for transaortic access, but the TAVI group will further be referred to as TA-TAVI. Depending on valve type, rapid pacing was performed both during valve dilation and prosthesis release. Approximately 50% of the patients were extubated immediately after the procedure and transferred to the ICU. Valve types used included five JenaValves (3 × 23 mm, 2 × 25 mm), one Medtronic EvolutR 23 mm, 46 Edwards Sapien XT/3 (1 × 23 mm, 16 × 26 mm, 1 × 29 mm)/(14 × 23 mm, 13 × 26 mm, 1 × 29 mm), in the SURD-AVR group 35 Perceval S (4 × S, 12 × M, 11 × L, 6 × XL) and 16 Edwards Intuity (1 × 19 mm, 3 × 21 mm, 7 × 23 mm, 4 × 25 mm, 1 × 27 mm). Erythrocytes, fresh-frozen plasma and platelet transfusions were administered if required.

### 2.3. Matching

The sutureless cohort was compared to a matched-pair group for statistical analysis in a retrospective and descriptive manner. Matched patients were also selected during the same time period out of 214 patients undergoing TA-TAVI or alternate access (e.g., transaortic). TF-TAVI patients were excluded. Age ± 5 years, sex, BMI ± 5, emergency indication, dialysis and additive EuroSCORE ± 5 were used for match-pair analysis (1:1). Thus, 52 patients were matched and compared. Since this is a descriptive analysis, endpoint parameters were defined as outcomes in terms of in-hospital mortality, adverse clinical events and mid-term survival.

### 2.4. Statistics

Continuous variables are shown as mean ± standard deviation or as median and range, categorical data are shown as percentages. To elaborate differences between both arms preoperative, operative and postoperative data were analyzed by the Student’s *t*-test, Fisher’s exact test and the Χ2 test. A two-tailed *p* value less than 0.05 was considered significant. SPSS 25.0 software (SPSS, Inc., Chicago, IL, USA) was used for all statistical analyses.

## 3. Results

### 3.1. Preoperative Data

The cohorts of both study arms showed comparable baseline characteristics, but differed in the presence of NYHA class III and IV (TAVI 83% vs. SURD-AVR 62%; *p* = 0.04). None of the patients underwent an emergency operation. Furthermore, no significant difference was found in terms of COPD, pulmonary hypertension, diabetes or peripheral vascular disease (PAD) (TAVI 29% vs. SURD-AVR 15%; *p* = 0.98) preoperatively. None of the patients required dialysis preoperatively. While no difference was found for LV function, nine patients presented with impaired LVF (TAVI 13% vs. SURD-AVR 4%; *p* = 0.160). Significantly more TAVI patients (44% vs. 17%; *p* = 0.003) underwent PCI with stenting within 6 months preoperatively. Baseline characteristics are depicted in Table 1. 

### 3.2. Operative Data

Operative data showed statistical differences between both groups, since the SURD-AVR group had significantly longer operation times (273 ± 122 vs. 203 ± 65 min; *p* < 0.001). Furthermore, intraoperative blood transfusion was significantly higher in the SURD-AVR group (1405 ± 1032 vs. 981 ± 770 mL; *p* = 0.014). Operative outcome data are displayed in Table 2. 

### 3.3. Postoperative Data

In-hospital death did not differ between both groups (TAVI: *n* = 3, 5.8% vs. SURD-AVR: *n* = 2, 3.8% death; *p* = 0.647). Both deceased SURD-AVR patients underwent concomitant CABG surgery. Moreover, Kaplan–Meier-estimated survival (Figure 1) after 12 months showed no significant difference between both procedures, but with a notable tendency in favor of SURD-AVR (TAVI: 73 ± 9% vs. SURD-AVR: 94 ± 3%; log rank *p* = 0.058).

Scarce significant differences between both groups were noted in the analysis of postoperative data (Table 2). The TAVI group showed higher maximum creatinine values (1.37 ± 0.88 vs. 1.0 ± 0.51 *p* = 0.027), and although more patients required hemofiltration the difference remained non-significant (*p* = 0.169). Interestingly, no difference in ventilation time (*p* = 0.914) and need of surgical re-exploration (TAVI 1.9% vs. SURD-AVR 0%; *p* = 0.315) was noted. The incidence of cardiac low-output syndrome was low in both groups *n* = 3; *p* = 1. During the postoperative course the need for red cell transfusion did not show any difference, albeit numerically higher in the TAVI cohort (TAVI 2.02 ± 3.84 units vs. SURD-AVR 1.14 ± 2.14 units; *p* = 0.151). AV block requiring pacemaker implantation was more frequent after SURD-AVR, although there was no significant statistical difference (*p* = 0.169). 

TAVI led to significantly higher rates of aortic regurgitation (AR) at discharge (*p* = 0.013). However, relevant paravalvular leakage was found in only one TAVI patient. All other regurgitations of TAVI patients were mild at most. In the SURD-AVR group only three patients showed trace AR within the prosthetic ring and no paravalvular leakage. 

## 4. Discussion

We present a match-paired analysis comparing two modern strategies for the operative treatment of patients with symptomatic aortic stenosis. In comparison to several recent reports and meta-analyses, we included TAVI procedures via TA access and one transaortic. Since indication for TAVI has extended towards intermediate- and low-risk cohorts, our aim was to include almost exclusively TA patients, as TA access is predominantly performed in patients at higher risk levels, who suffer multiple co-morbidities [23,24]. Moreover, TA implantation requires full anesthesia and a surgical access site, making the procedure more comparable to surgical valve replacement. Noteworthy is the relatively low incidence of PAD in our TA-TAVI cohort, although PAD is the chief cause for conversion to TA or alternate access. However, our center prefers TA access over TF if groin vessels measure below 5–7 mm in diameter, show eccentric calcification or extensive tortuosity, avoiding the risk of peripheral vessel complication.

Our short-term results prove the safety and efficacy of both TA-TAVI and SURD-AVR procedures in this intermediate- to high-risk cohort. In comparison to previous studies, our SURD-AVR cohort presented with more co-morbidities resulting in higher ES, as SURD-AVR has been predominantly chosen for patients with increased operative risk and need for concomitant procedures in our center. This may explain a slightly higher in-hospital mortality rate as compared to other reports [13,15,17,18,19]. The Sutureless and Rapid-Deployment Aortic Valve Replacement International Registry (SURD-IR) data reported an overall hospital mortality of 2.1% with a logistic ES of 11.3  ±  9.7% in 3343 patients [25], which is markedly lower than in our SURD-AVR cohort. The comparability of the cohort is even more emphasized due to the fact that 46% of the SURD-AVR group underwent concomitant CABG and 44% of the TAVI patients underwent staged PCI prior to surgery. In-hospital mortality was comparable in this subpopulation and coronary artery disease (CAD) treatment had no negative impact on outcomes. The combined treatment of symptomatic aortic stenosis and CAD must be the topic of future investigations comparing both treatment options. 

Moreover, postoperative neurological complications including stroke (2.8%) and transient ischemic attack (1.1%), as reported in the SURD-IR registry data [25], and a stroke rate of 2.7% in TA-TAVI procedures of the PARTNER 1 cohort [26] were not observed in the TAVI group, although we did not use embolic protection systems during TAVI. Despite the occurrence of one cerebrovascular accident in the SURD-Valve group, no statistical significance resulted. Our favorable neurological results might be related to the fact that antegrade wire placement is associated with a reduced risk of manipulation of calcified valves compared to the retrograde approach used in TF access. Recent data, however, reported similar stroke rates in TF and TA salvage procedures, thus not confirming this hypothesis [23,27]. Moreover, we found only a small number of paravalvular leakages, especially after SURD-AVR. In our experience, it is of paramount importance to avoid implantation of these types of prostheses in bicuspid valves or oval aortic annulae.

Despite the common incidence of atrioventricular block requiring permanent pacemaker implantation in both procedures, our cohort had a very low incidence of complete AV block requiring permanent pacemaker implantation, especially after TAVI [20,25,28]. This may be related to our implantation technique and valve choice. Most of our TAVI patients received balloon-expandable Edwards Sapien valves, which reportedly cause less AV block [11,29]. Since approval and use of the Sapien 3 prosthesis, we did not perform balloon-dilation of the aortic valve before balloon-expansion of the prosthesis. Moreover, our heart team’s implantation technique tends to place the valves slightly towards the coronary artery, which could extrude the native calcified cusps less towards the region of minor resistance, causing structural damage and edema within the conducting system (AV node and left bundle branches). 

The intraoperative need for blood transfusion was obviously higher in the SURD-AVR group, as reported in in a recent meta-analysis due to procedural reasons [30]. While blood transfusion is controversially discussed, it does not seem to have a negative impact on outcomes as reported by Bjursten et al. after conventional valve replacement operations [31]. Higher rates of renal failure requiring dialysis in the TAVI group was not significant and may also be related to preoperatively elevated creatinine levels in the TAVI group. 

Despite our comparable results, our study includes several limitations, first and foremost due to its retrospective design and small cohort. Moreover, the procedures differ relevantly due to mandatory use of cardiopulmonary bypass in the SURD-AVR group. We do not offer invasive pressure gradients, which further assess the procedural success quantitatively. Alternative access sites such as trans-axillary and trans-carotid, may obtain beneficial outcomes over TA but were not included in this study due to insufficient case count.

Clinical follow-up results and quality of life assessments are missing. The SURD-AVR group includes concomitant CABG procedures, while TAVI patients did undergo PCI prior to surgery. 

In conclusion, our present data underline the safety and efficacy of SURD-AVR compared to TA-TAVI in patients at intermediate- to high-risk for conventional surgery. Combining the advantage of standard diseased-valve removal with shorter procedural times, sutureless aortic valve replacement may be considered as an alternative treatment for high-risk patients considered to be in the “gray zone” between TAVI and conventional surgery, especially if concomitant myocardial revascularization is required. However, prospective analyses and long-term follow-up data are necessary to further prove these findings and develop future implications of alternate treatments for intermediate- and high-risk patients with symptomatic aortic stenosis.

## Figures and Tables

**Figure 1 jcm-10-03072-f001:**
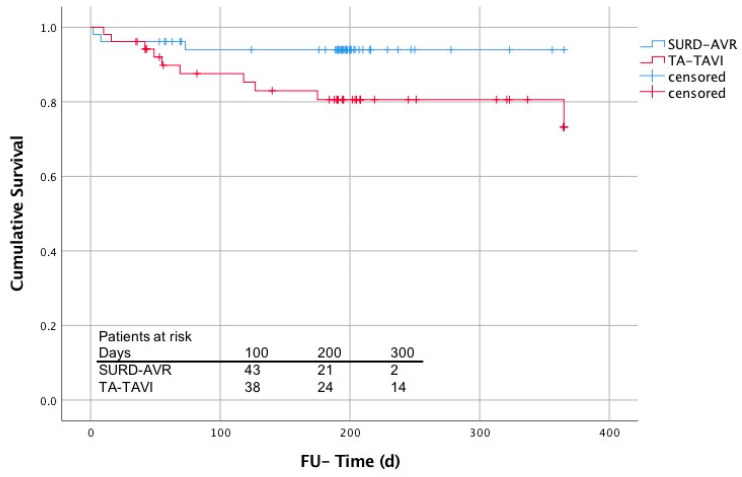
Estimated survival (Kaplan–Meier) comparing SURD-TAVI and TA-TAVI.

**Table 1 jcm-10-03072-t001:** Baseline characteristics.

*n* = 52 Each Group	TAVI (*n* and Perc. or Mean ± SDM)	SURD-AVR (*n* and Perc. or Mean ± SDM)	*p* Value
female	32 (62%)	32 (62%)	
age, year	77 ± 4.3	75 ± 4.0	0.051
log. EuroScore I	19 ± 12	17 ± 10	0.257
STS Score	4.48 ± 2.76	3.93 ± 2.59	0.339
art. hypertension	45 (87%)	45 (87%)	
hyperlipidemia	28 (54%)	28 (54%)	
impaired LVF	7 (13%)	2 (4%)	0.160
renal insufficiency	16 (31%)	9 (17%)	0.117
previous MI	8 (15%)	5 (9.6%)	0.371
COPD	18 (35%)	15 (29%)	0.527
PAD	15 (29%)	8 (15%)	0.98
NYHA I-IV	2.87 ± 0.44	2.67 ± 0.59	0.062
emergency indication	0 (0%)	0 (0%)	
creatinine (mg/dL)	1.29 ± 0.73	0.92 ± 0.40	0.002
bilirubin (mg/dL)	0.56 ± 0.35	0.62 ± 0.28	0.416
BMI	27.3 ± 5	28 ± 5	0.440
diabetes	21 (40%)	22 (42%)	0.971
dialysis	0 (0%)	0 (0%)	

LVF, left ventricular function; MI, myocardial infarction; COPD, chronic obstructive pulmonary disease; PAD, peripheral arterial disease; NYHA, New York Heart Association classification; and BMI, body mass index.

**Table 2 jcm-10-03072-t002:** Operative and postoperative data.

*n* = 52 Each Group	TAVI (*n* and Perc. or Mean ± SDM)	SURD-AVR (*n* and Perc. or Mean ± SDM)	*p* Value
blood transfusion (U)	0.72	1.46	0.014
CV accident/stroke	0 (0%)	1 (1.9%)	0.315
max. bilirubin (mg/dl)	0.85 ± 0.5	0.5 ± 0.7	0.115
complete AV block	1 (1.9%)	4 (7.7%)	0.169
RF requiring dialysis	4 (7.7%)	1 (1.9%)	0.169
ventilation time (h)	26 ± 66	25 ± 21	0.914
re-intubation	4 (7.7%)	1 (1.9%)	0.169
postoperative blood transfusion	2.02 ± 3.84	1.14 ± 2.14	0.151
low-output syndrome	3 (5.8%)	3 (5.8%)	1
AR at discharge			
-trace to mild	13 (25%)	3 (5.8%)	
-moderate	1 (1.9%)	0 (0%)	0.013
re-thoracotomy	1 (1.9%)	0 (0%)	0.315

CV, cerebrovascular; AV, atrioventricular; RF, renal failure; and AR, aortic regurgitation.

## Data Availability

All relevant data are included within the manuscript. All other relevant data presented in this study are available on request from the corresponding author.

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
