# Peer review of "Are Sutureless and Rapid-Deployment Aortic Valves a Serious Alternative to TA-TAVI? A Matched-Pairs Analysis"

_jcm, 2021, doi:10.3390/jcm10143072_

Round 1

Reviewer 1 Report

This study describes two surgical strategies for the treatment of severe AS in patients not suitable for trans-femoral TAVI.

This is a small matched cohort comparing SAVR using a suture-less valve vs. trans-apical TAVI

The conclusion is that the outcome is similar and suture-less valve should be the default treatment in these patients

Comments:

  1. English should be corrected throughout the manuscript
  2. The fact the multivariable regression analysis did not reveal any predictor is due to the small number of patients and it should be omitted from the manuscript.
  3. Please provide more details on the procedure - valves sizes etc...  
  4. Please provide STS score
  5. Please discuss other alternative non-invasive and nor surgical approaches such as trans-axillary, trans-carotid, and trans-caval
  6. How do you explain the excess 1y mortality rate of 20% in trans-apical TAVI

Author Response

This study describes two surgical strategies for the treatment of severe AS in patients not suitable for trans-femoral TAVI.

This is a small matched cohort comparing SAVR using a suture-less valve vs. trans-apical TAVI

The conclusion is that the outcome is similar and suture-less valve should be the default treatment in these patients

Comments:

  1. English should be corrected throughout the manuscript

We would like to thank the reviewer for his recommendation. The manuscript has been checked and corrected by a native English-speaking colleague.

  1. The fact the multivariable regression analysis did not reveal any predictor is due to the small number of patients and it should be omitted from the manuscript.

We would like to thank the reviewer for this suggestion and agree that the sample size is indeed too small for a proper regression analysis. Thus, regression analysis has been omitted entirely from the manuscript.

  1. Please provide more details on the procedure - valves sizes etc...  

Thank you for your suggestion. We added valve sizes to the manuscript.

  1. Please provide STS score

STS Scores have been included and are shown in table 1.

  1. Please discuss other alternative non-invasive and nor surgical approaches such as trans-axillary, trans-carotid, and trans-caval

We included a statement within the limitations section. However, we agree that theses alternative access sites seem to provide beneficial outcomes over TA. We still observe frequent calcification of supra-aortic branches, which often hinder these approaches. Additionally, we do not use trans-carotid approaches at our center. The trans-axillary approach has also not been used frequently since TA results are good and many patients present with kinking, calcification, and stiffness of supra-aortic branches. Moreover, patients often present with s/p CABG. In those cases, we do not want to put a patent in-situ mammary bypass at risk. Nevertheless, alternative approach are expanding and we also try to support the trans-subclavian/axillary approach, which offers many advantages over TA (often no intubation needed, no thoracotomy, use of self-expandable valves for heavily calcified annulus, …).

  1. How do you explain the excess 1y mortality rate of 20% in trans-apical TAVI

In addition to the morbidity reflected by STS Score and EuroSCORE, the TAVI cohort may present with frailty un-objectifiable and not fully depicted in risk calculators. This high-risk cohort certainly includes patients with a limited prognosis of 2y or lower which may explain the worse outcomes after 1 year.

Reviewer 2 Report

This is a well written manuscript addressing an important topic in the management of high risk patients with aortic valve stenosis. However, some additional concerns need to be addressed.

  1. In abstract, the authors stated “This present paired-match analysis aims to compare patients undergoing transcatheter aortic valve implantation to Sutureless and rapid-deployment aortic valve Implantation”. What are the outcome points? Even in methods in abstract as well as in main text these information are missing.
  2. The authors conducted 1.1 matching, information about the propensity caliper should be given. In addition, to assess the quality of matching, the absolute standardized mean should be reported in Table 1 comparing baseline characteristic instead of p-value.
  3. In results in abstract, the authors stated “In-hospital death was comparable between transcatheter aortic valve implantation (mean age 77 ± 4.3 years) and Sutureless and rapid-deployment aortic valves Implantation groups (mean age 75 ± 4.0 years) including 32 females in each group (5.8% vs. 1.9%; P=0.308).”

This sentence is confusing, the authors started reporting about in-hospital mortality, but did not mention its rates, instead they reported about the mean age in each group.

  1. In abstract, the authors mentioned that “Complete heart block requiring permanent pacemaker was relatively rare in both groups (1.9% vs. 7.7%; P=0.169)”. I would not consider 7.7% as a rare incidence.
  2. In the conclusion, the authors stated “Since Suture-less and rapid-deployment aortic valve implantation is as safe and effective as transapical transcath-eter aortic valve implantation, combining the advantage of standard diseased valve removal with shorter procedural times, Sutureless and rapid-deployment aortic valve replacement may be the first-line treatment for patients with elevated operative risk considered in the "gray zone" between transcatheter aortic valve implantation and conventional surgery especially if concomitant myocardial revascularization is required.”
  3. I consider the conclusion that Sutureless and rapid-deployment aortic valve replacement may be the first-line treatment for patients with elevated operative risk considered in the "gray zone" between transcatheter aortic valve implantation and conventional surgery as not supported with evidence from the current study. There was no conventional surgery group (received a conventional stented aortic valve replacement) included to draw such a conclusion. It is even not mentioned in this study what were the safety and efficacy endpoints. One can argue that pateitns who underwent conventional AVR with stented valve would had similar outcome as rapid deployment or sutureless valves.
  4. In Table 1: please present the number of patients and not only the percentages.

The authors stated that “In SURD-AVR group only 3 patients showed trace AR within the prosthetic ring and no paravalvular leakage”. I would to congratulate the authors for the very low incidence of paravalvular leakage in their patients, which is remarkably lower than those reported from other studies. I would suggest this point to be commented in the discussion. It will be interesting if the authors mention some words on how to achieve such a low rate of paravalvular leakage. 

Author Response

This is a well written manuscript addressing an important topic in the management of high risk patients with aortic valve stenosis. However, some additional concerns need to be addressed.

We would like to thank reviewer #2 for his careful evaluation of our manuscript. We have heeded his suggestions and performed the necessary revisions where possible as recommended.

  1. In abstract, the authors stated “This present paired-match analysis aims to compare patients undergoing transcatheter aortic valve implantation to Sutureless and rapid-deployment aortic valve Implantation”. What are the outcome points? Even in methods in abstract as well as in main text these information are missing.

Thank you for your comment. Here we present a descriptive comparison of both procedures focus on the in-hospital outcome and procedural outcomes. Therefore, we did not defined primary and secondary outcome parameters. We added this information to the method section.

  1. The authors conducted 1.1 matching, information about the propensity caliper should be given. In addition, to assess the quality of matching, the absolute standardized mean should be reported in Table 1 comparing baseline characteristic instead of p-value.

We did not perform a propensity matching due to the sample size and comparability of the cohorts. This is a mere match-pair analysis in which Age ± 5 years, sex, BMI ± 5, emergency indication, dialysis and additive EuroSCORE ± 5 were used for match-pair analysis.

  1. In results in abstract, the authors stated “In-hospital death was comparable between transcatheter aortic valve implantation (mean age 77 ± 4.3 years) and Sutureless and rapid-deployment aortic valves Implantation groups (mean age 75 ± 4.0 years) including 32 females in each group (5.8% vs. 1.9%; P=0.308).”

This sentence is confusing, the authors started reporting about in-hospital mortality, but did not mention its rates, instead they reported about the mean age in each group.

Thank you for this observation. We changed the position of the mortality rates to avoid confusion.

  1. In abstract, the authors mentioned that “Complete heart block requiring permanent pacemaker was relatively rare in both groups (1.9% vs. 7.7%; P=0.169)”. I would not consider 7.7% as a rare incidence.

We absolutely agree with this interpretation and changed the statement within the manuscript.

  1. In the conclusion, the authors stated “Since Suture-less and rapid-deployment aortic valve implantation is as safe and effective as transapical transcath-eter aortic valve implantation, combining the advantage of standard diseased valve removal with shorter procedural times, Sutureless and rapid-deployment aortic valve replacement may be the first-line treatment for patients with elevated operative risk considered in the "gray zone" between transcatheter aortic valve implantation and conventional surgery especially if concomitant myocardial revascularization is required.”
  2. I consider the conclusion that Sutureless and rapid-deployment aortic valve replacement may be the first-line treatment for patients with elevated operative risk considered in the "gray zone" between transcatheter aortic valve implantation and conventional surgery as not supported with evidence from the current study. There was no conventional surgery group (received a conventional stented aortic valve replacement) included to draw such a conclusion. It is even not mentioned in this study what were the safety and efficacy endpoints. One can argue that pateitns who underwent conventional AVR with stented valve would had similar outcome as rapid deployment or sutureless valves.

The conclusion of the manuscript may have been favouring one treatment method, while we in fact agree with the reviewer’s opinion. Hence amendments to the conclusion were made to underline the comparability of the procedure in this patients’ cohort, making it a useful treatment method for selected patients.

  1. In Table 1: please present the number of patients and not only the percentages.

We included the number of patients to the percentages in the tables.

The authors stated that “In SURD-AVR group only 3 patients showed trace AR within the prosthetic ring and no paravalvular leakage”. I would to congratulate the authors for the very low incidence of paravalvular leakage in their patients, which is remarkably lower than those reported from other studies. I would suggest this point to be commented in the discussion. It will be interesting if the authors mention some words on how to achieve such a low rate of paravalvular leakage. 

Thank you for this comment. Out premise is to accurately size these special valve types to ensure optimal position and do not accept relevant leakage. We are strict in terms of contra-indication and do not implant these prostheses in bicuspid valves or significant oval aortic annulae. On the non-coronary sinus additional sutures may be placed, if the sinus is significantly larger than the other sinus to avoid dislocation if an Intuity valve is used.

Reviewer 3 Report

I thank the authors for the well-described match-paired analysis comparing TA-TAVI and SURD-AVR.

I have a few comments.

TAVI is not only feasible in intermediate and high-risk patients. Latest data show that TAVI seems to be also preferable in low-risk patients (Mack MJ et al., N Engl J Med 2019; Popmaa JJ et al., N Engl J Med 2019). Please revise this point of the introduction.

Why did you chose the transapical approach for your selected patients - please provide detailed information. Due to the better outcome after TF-TAVI, the transapical access is only choosen, if the transaortic access is not possible due to calcifications. But only 29% of your TAVI-patients had PAD?

In this context: your description, that the transapical approach is predominantly performed in patients with higher risk levels is not state-of-the art. A transaortic approach is preferred whenever possible. In case of to small groin vessels (your diameter as limitation for TF seems to be to strict) the subclavian approach is mostly possible. So, what is the impact on daily practice and the novelity of your study? 

See the comments above and your study design - you cannot state, that "SURD-AVR may be the first line treatment for high risk patients considered in the gray zone between TAVI and conventional surgery". You can only state, it is safety compared to TA-TAVI. 

Results:
- Please clarify the description of your results. In some party it is not recognizable which values in brackets refer to which group
- Please check wether all abbreviations in the tables are also explained
- The rate of pacemaker implantations post-TAVI seems very low - even for Sapien prosthesis. What are your criteria for a pacemaker implantation after TAVI? How many new conduction disturbances occurred?
-The TAVI group showed a higher Creatinine value after the procedure but the baseline value was also significantly higher in this group

Author Response

I thank the authors for the well-described match-paired analysis comparing TA-TAVI and SURD-AVR.

I have a few comments.

We would like to thank reviewer #3 for his meticulous review and productive suggestions.

TAVI is not only feasible in intermediate and high-risk patients. Latest data show that TAVI seems to be also preferable in low-risk patients (Mack MJ et al., N Engl J Med 2019; Popmaa JJ et al., N Engl J Med 2019). Please revise this point of the introduction.

We agree with the observation of the reviewer. Hence, we revised our introduction and added the proper reference.

Why did you chose the transapical approach for your selected patients - please provide detailed information. Due to the better outcome after TF-TAVI, the transapical access is only choosen, if the transaortic access is not possible due to calcifications. But only 29% of your TAVI-patients had PAD?

Thank you for your comment. The relatively small percentage of PAD in the TAVI-group was also noted by our analysis. We believe this to be a possible matching phenomenon since PAD is one of the most common reasons for TA approaches. Moreover, in this study period the transfemoral approach was less aggressive and advanced, since sheath sizes for TF access have been larger and indication for TA was also based on arterial diameters, tortuosity of the aorta and iliacal artery and not only on calcification.

In this context: your description, that the transapical approach is predominantly performed in patients with higher risk levels is not state-of-the art. A transaortic approach is preferred whenever possible. In case of to small groin vessels (your diameter as limitation for TF seems to be to strict) the subclavian approach is mostly possible. So, what is the impact on daily practice and the novelity of your study? 

We share the reviewer’s opinion, however due to the retrospective character of the study, the data is susceptible to its inherit drawbacks. Furthermore, alternate approaches are not always technically feasible and not suitable for all valve prosthesis types, which in our opinion should be primarily depended on annular measurement. Furthermore, there are recurrent cases of patients, who have to be converted to full sternotomy and surgical aortic valve surgery with help of cardiopulmonary bypass, due to hemodynamic instability, TAVI procedure complications or re-operations for TAVI prosthesis endocarditis and show surprisingly acceptable outcomes. This study compares a TAVI cohort, which is clearly a high-risk cohort and without doubt eligible for TAVI. Nevertheless, as mentioned above, in some cases (concomitant CABG, heavily calcified annulus and LVOT, etc.) surgical approach may be inevitable. SURD-AVR represents a possible compromise for these patients with acceptable outcomes.

See the comments above and your study design - you cannot state, that "SURD-AVR may be the first line treatment for high risk patients considered in the gray zone between TAVI and conventional surgery". You can only state, it is safety compared to TA-TAVI. 

We agree with the reviewer’s opinion, and we therefore modified the conclusion of the manuscript accordingly.

Results:
- Please clarify the description of your results. In some party it is not recognizable which values in brackets refer to which group

Thank you for your comment. We addressed this issue.

- Please check wether all abbreviations in the tables are also explained

Thank you for your detailed review. All abbreviations are now explained.

- The rate of pacemaker implantations post-TAVI seems very low - even for Sapien prosthesis. What are your criteria for a pacemaker implantation after TAVI? How many new conduction disturbances occurred?

Thank you for your observation. Indeed, we do observe only few conduction disturbances necessitating perioperative pacemaker implantation. We are currently writing a manuscript with focus on this issue and observe a risk for relevant AVB after TA-TAVI of ca. 5%. We predominantly have used and do use Sapien 3 and Sapien 3 Ultra valves. We try not to oversize and tend to implant our valves slightly towards the aortic site; away from conduction structures which may influence these results.

-The TAVI group showed a higher Creatinine value after the procedure but the baseline value was also significantly higher in this group

Thank you for your comment. This issue has been addresses within the discussion section.

Round 2

Reviewer 1 Report

The revised manuscript is acceptable for publication

Reviewer 2 Report

After changes made by the authors  in the manuscript, it  has beensufficiently improved to warrant publication in JCM.

Reviewer 3 Report

I thank the authors for improving their manuscript.